# Impact of Constraint-Induced Movement Therapy (CIMT) on Functional Ambulation in Stroke Patients—A Systematic Review and Meta-Analysis

**DOI:** 10.3390/ijerph191912809

**Published:** 2022-10-06

**Authors:** Ravi Shankar Reddy, Kumar Gular, Snehil Dixit, Praveen Kumar Kandakurti, Jaya Shanker Tedla, Ajay Prashad Gautam, Devika Rani Sangadala

**Affiliations:** 1Department of Medical Rehabilitation Sciences, College of Applied Medical Sciences, King Khalid University, Abha 61421, Saudi Arabia; 2College of Health Sciences, Gulf Medical University, Ajman 4184, United Arab Emirates

**Keywords:** lower-extremity CIMT, stroke, gait speed, balance, cardiovascular, functional ambulation

## Abstract

Constraint-induced movement therapy (CIMT) has been delivered in the stroke population to improve lower-extremity functions. However, its efficacy on prime components of functional ambulation, such as gait speed, balance, and cardiovascular outcomes, is ambiguous. The present review aims to delineate the effect of various lower-extremity CIMT (LECIMT) protocols on gait speed, balance, and cardiovascular outcomes. Material and methods: The databases used to collect relevant articles were EBSCO, PubMed, PEDro, Science Direct, Scopus, MEDLINE, CINAHL, and Web of Science. For this analysis, clinical trials involving stroke populations in different stages of recovery, >18 years old, and treated with LECIMT were considered. Only ten studies were included in this review, as they fulfilled the inclusion criteria. The effect of CIMT on gait speed and balance outcomes was accomplished using a random or fixed-effect model. CIMT, when compared to controlled interventions, showed superior or similar effects. The effect of LECIMT on gait speed and balance were non-significant, with mean differences (SMDs) of 0.13 and 4.94 and at 95% confidence intervals (Cis) of (−0.18–0.44) and (−2.48–12.37), respectively. In this meta-analysis, we observed that despite the fact that several trials claimed the efficacy of LECIMT in improving lower-extremity functions, gait speed and balance did not demonstrate a significant effect size favoring LECIMT. Therefore, CIMT treatment protocols should consider the patient’s functional requirements, cardinal principles of CIMT, and cardiorespiratory parameters.

## 1. Introduction

Stroke is the second leading cause of death, and its rising incidence, mortality, and disability impose a significant global burden [1]. Stroke is pronounced in low- and middle-income countries, diverse age groups, and gender [2]. Six and twelve months following a stroke, the mortality and disability rates increase dramatically (55.9% and 61.0%, respectively) [3]. The number of new cases, the prevalence of functional disability, and the mortality rate associated with stroke indicate the need for improved rehabilitation techniques that reduce the time to recovery [3].

Home- and community-based ambulation are important rehabilitation goals for post-stroke patients [3,4,5]. Inter-limb coordination, proprioception, muscle strength, static and functional balance, gait speed, stance phase of the non-paretic leg, and cardiovascular fitness showed a strong relationship with walking ability among post-stroke participants [6,7,8,9,10]. The gait parameters of post-stroke patients are affected by sensorimotor impairments, muscle paresis, impaired proprioception, and motor control [11,12,13,14,15]. Among the gait parameters, gait speed has shown a prominent association with other spatiotemporal, kinematic, and kinetic gait variables [15].

Physical inactivity leads to vascular impairments and reduction in cardiovascular fitness among post-stroke subjects, which, if not improved, results in recurrent strokes [16,17,18]. Following a stroke, cardiorespiratory fitness and the vascular system are severely compromised [18,19,20,21]. The neurological impairments and loss in cardiorespiratory endurance and blood supply to the muscles lead to physical intolerance and higher energy expenditure [17,22], while the autonomic dysregulation that follows a stroke leads to aberrant blood pressure and uncontrolled heart rate [23]. The disparities observed in the motor recovery process, balance, muscle strength, and confidence levels are the factors for a wide range of gait speeds (0.76–1.09 m/s) in people who have suffered from a stroke [24,25].

In the first month following a stroke, muscle weakness and incoordination determine gait functions, whereas cardiorespiratory endurance substantially influences gait results after one month [26]. Determinants of walking ability are usually outlined by factors such as cardiorespiratory endurance, mobility, and balance. The parameters for walking ability govern goal achievement and community participation in chronic stroke patients. Moreover, a reasonable relationship exists between cardiovascular endurance, walking parameters, and balance with the distinguishable enhancement of walking ability in post-stroke subjects [6,8,26,27,28].

Constraint-induced movement therapy (CIMT) proved to be an effective rehabilitation tool in remedying several impairments encountered by the stroke population [29,30,31,32,33]. The previous success of CIMT as a treatment strategy for upper-limb rehabilitation has inspired researchers to further investigate its utility in lower-limb rehabilitation among the stroke population [34,35]. The application of constraint to the unaffected limb enhanced the use of the affected limb while performing functional activities significantly but failed in achieving skillful activities [36]. Restraining the unaffected extremity and intensive training of the affected extremity in various functional-oriented tasks emphasizing repeated practice and task shaping improved both paretic limb participation and the ability to perform daily activities [37]. The enhancement and commitment of affected limb participation in day-to-day functional activities attained through the continuous change in the behavior of internal properties of a task are called a “transfer package” [38]. The efficient application and modification of restraint to the unaffected extremity, type of functional task and its practice, and shaping and behavior of internal properties of the task will yield purposeful and meaningful results. The shaping technique comprises identifying the task and providing ideal feedback to attain the task, followed by the gradual increase in the task’s difficulty. Restraint of the non-paretic extremity used in unilateral motor deficits cannot be applied to the lower extremities because humans are predominantly bipedal. Such restraint may affect gait symmetry, speed, and inter-limb coordination [39,40]. The shift in the paradigm from learned non-use to learned misuse has removed the barriers to applying CIMT in bilateral motor deficits and lower-limb training [37,40]. The use-dependent cortical reorganization promoted through emphasis and repetitive practice of the paretic limb discouraged non-paretic restraint in CIMT training [41]. Thus, intensive exercise with more repetitions, task shaping, and transfer package remained an essential component of lower-extremity CIMT (LECIMT) training. Previous reviews on gait training approaches in improving independent ambulation recommended intensive and repeated practice, individually tailored functional tasks, task shaping, and behavioral strategies. Expert panel recommendations and neuroimaging evidence suggest a rationale for the applicability of LECIMT in post-stroke subjects [42,43,44]. However, the flexible nature of CIMT allowed researchers to incorporate all or some of the components into LECIMT. Recent studies on LECIMT reported significant improvements in gait parameters and balance and traceable effects on functional mobility, ambulation, motor functions, and cardiovascular parameters. 

Even though CIMT is a suitable treatment approach for improving upper extremity functions, its effects on the lower extremities are debatable due to ambiguity in applying integral components of CIMT and a lack of well-designed protocols. Therefore, the current review and meta-analysis were conducted to substantiate and create evidence of various LECIMT protocols and their effects on gait speed, balance, and cardiovascular parameters, which were prime elements contributing to functional ambulation.

## 2. Materials and Methods

### 2.1. Selection Criteria of Studies for This Review

RCTs that compared lower-limb training utilizing the principles of CIMT with conventional therapy or other neurorehabilitation techniques were considered for this review. Randomized controlled trials (RCTs), which included first-time or recurrent, hemorrhagic or ischemic post-stroke subjects aged above 18 years, were considered. Studies that measured and analyzed the gait speed, balance, and cardiovascular parameters were primarily included in the review (Protocol registered under Prospero database with registration no: CRD42021260203).

### 2.2. Literature Search and Study Selection

Two independent reviewers (DRS and RSR) searched for pertinent English-language articles published between 2000 and 2022. In the initial phase, one reviewer (KG) reviewed recognized titles and references for their relevance. Later, studies were screened based on the selection criteria for their relevance. Finally, the expert reviewer (PKK) resolved the differences in the studies.

We searched the databases PEDro, Web of Science, PubMed, Scopus, EBSCO, MEDLINE, CINHAL, and Science Direct for relevant publications. The MeSH keywords included: stroke (chronic, subacute, acute), CIMT, forced use, restricted limb or extremity cerebrovascular accidents, hemiparesis, hemiplegia, gait speed, cardiovascular, balance, blood pressure, and percentage of heart rate maximum. Table 1 provides a summary of the search strategy’s specifics.

### 2.3. Data Extraction

Two independent reviewers (SD and APG) were involved in the process of collecting relevant data from the included studies. The third reviewer (PKK) was invited to sort out discrepancies between the two reviewers in authenticating the data. The information extracted from the studies was based on 1. patients’ features, 2. PEDro score, 3. sample size, 4. eligibility standards, 5. outcome dimensions, and 6. constraint applications.

### 2.4. Evaluation of Methodological Quality and Level of Evidence

PEDro scale, which consists of 10 domains, was used to assess the methodological quality of included studies. RCTs with a score of 9–10 on the PEDro scale were assigned as “excellent”, 6–8 as “good”, 4–5 as “fair”, and less than four as “poor” methodological quality. Two independent assessors (JST and SD) used the PEDro scoring to evaluate the methodological quality of the studies. The third assessor was contacted to resolve the disparity in assigning the score. 

Further to the criteria mentioned above, high-quality RCTs with PEDro ≥ 6 are considered as level 1 evidence and have been divided into two subcategories, level 1a (>1 study with PEDro ≥ 6) and level 1b (1 study with PEDro ≥ 6). Low-quality RCTs with PEDro < 6 designated as Level 2 evidence. This categorization was imbibed from Sackett et al.’s criteria outlined by Dixit and Gular [45]. Outcome measures in the current study were categorized as levels 1a, 1b, and 2 based on the number of quality trials supporting it.

### 2.5. Risk of Bias Assessment

Review manager 5.4.1 software (London, UK) was utilized to evaluate and synthesize the risk of bias among included studies. Two reviewers (KG and SD) independently rated individual studies on the following domains: 1. allocation concealment, 2. random sequence generation, 3. blinding of participants, 4. blinding of outcome data, 5. incomplete outcome data, 6. selective reporting, 7. discrepancy of intervention between the group, and 8. other biases. Usually, descriptors such as high risk, low risk, and unclear were used to report the domains. An expert reviewer (RSR) was invited to resolve the variance of opinion between the reviewers in designating the descriptors.

### 2.6. Data Synthesis 

Meta-analysis was performed to determine the effect size of CIMT on gait speed, balance, and cardiovascular outcome measures, which were observed in the included studies. An outcome measure assessed in 2 or more trials was considered for meta-analysis. When descriptive values were reported as the median and range in the included studies, the mean and standard deviations were calculated using the conversion formulae. If data were not reported, the relevant authors were contacted through email and requested to provide the information. Review manager 5.4.1 software was used to conduct meta-analysis. 

The heterogeneity analysis (I2 statistics) was performed. Statistics were generated for the treatment effects on outcome measures if we found no clinical heterogeneity regarding the included subjects’ characteristics. The present meta-analysis considered I2 statistics with more than 50% value as considerable heterogeneity [46]. A random-effects model was used for analysis to obtain results. A pooled standardized mean difference (SMD) was calculated for outcome measures. 

## 3. Results

### 3.1. Search Results

Ten studies were included in the review process. Details regarding the process of database searches and exclusion and inclusion of studies are provided in Figure 1.

### 3.2. Characteristics of Included Studies

Among the ten studies, 329 subjects participated in the trials (212 males and 117 females). The stroke controlled-trial population consisted of middle-aged and elderly adults aged 40 to 70 years. All participants involved in the trials suffered either ischemic or hemorrhagic strokes. Considering the mean and standard deviation for the stroke duration throughout the analyzed studies, one study included subacute strokes [47], six studies included subacute/chronic [48,49,50,51,52,53], and three studies included chronic stroke population [54,55,56] ranging from 1.5 to 6.5 years. 

All included RCTs compared LECIMT with conventional physiotherapy or neurorehabilitation techniques. Six studies compared LECIMT with conventional physiotherapy [48,50,51,53,54,55], one study compared LECIMT with neuro-developmental therapy (NDT) [42], and three studies compared conventional physiotherapy as an adjunct to LECIMT with conventional physiotherapy [50,52,56,57]. The type of constraint utilized in the controlled trials was either restraint of a non-paretic lower extremity or augmentation of a paretic lower extremity. Restraining of a non-paretic lower extremity was provided by applying an ankle mass in five studies [50,51,53,54], immobilization of the knee in addition to a shoe insert in one study [56], and a negative kinematic restraint applied to a non-paretic lower extremity induced by robotic-assisted gait training (RAGT) in one study [55]. The mode of augmentation delivered to a paretic lower extremity among three clinical trials was diverse, with a compelling bodyweight shift to the paretic side (CBWS) [56], auditory feedback using a cane on the unaffected side [52], and enhancing the practice of the affected lower limb in training sessions and daily activities [58]. The total duration of constraint application was 20 min in two studies, 4.5 h in four studies, 90% of waking hours in one study, and applying while performing daily living activities in another study. 

The duration of intervention in the clinical trials ranged from 20 min to 55 h. Two clinical trials observed the immediate effects of single-session LECIMT intervention (20 min) [54,55]. However, in seven studies, the participants were intervened for less than 20 h, and in three studies, the intervention was delivered for more than 20 h. A diverse duration of intervention was observed among the studies with the single session to 32 sessions, and each session ranged from 20 min to 2 h 45 min per session. Table 2 summarizes the details of the included subjects, intervention, and outcomes.

### 3.3. Outcome Measures

Gait speed was pronouncedly observed in eight studies; compared to the control group, gait speed in the experimental group improved significantly in four trials [52,56,58,59] and showed similar improvements in four trials [48,50,54,55]. Berg balance scale in the experimental group showed marked improvements in two studies [56,59] and similar effects as a control group in one study [53]. Cardiovascular parameters, blood pressure, and percentage of heart rate maximum were reported in one study with the CIMT group showing no changes between pre- and post-sessions [50].

### 3.4. Methodological Quality and Level of Evidence

Among the comprised studies, three studies scored four [55,56,59], two studies scored five [47,50], one study scored seven [52], and three studies scored eight [50,51,53] on the Pedro scoring system. Therefore, based on the assessment, five studies were rated as fair, and five were rated as good. Details of the Pedro scores attained by the comprised studies are provided in Table 3. The gait speed and balance are supported by level 1b evidence, whereas cardiovascular outcomes with level two evidence. The level of evidence LECIMT on outcomes is summarized in Table 4.

The scores of the risk-of-bias assessment revealed that 50% of studies showed a high risk for concealed allocation and detection bias. In total, 75% of studies suffered from attrition bias and 100% of studies from participant bias. The included studies scored low risk for reporting bias and treatment imbalance. Details of the risk-of-bias assessment are given in Figure 2.

### 3.5. Quantitative Synthesis

Due to their single session, two of the ten included clinical trials were excluded from the analysis. A meta-analysis was performed for gait speed and balance because these factors were observed in six and three RCT studies, respectively. 

The CIMT group could not display a significant difference in gait speed compared to the control groups; the standard mean difference (SMD) was 0.13 at a 95% CI = −0.18–0.44, *p* = 0.42, with heterogeneity among the studies of I^2^ = 4% at *p* = 0.39. Furthermore, on post-follow-up, the CIMT group was unable to display a significant difference in gait speed when compared to the control groups (SMD = 0.32 at 95% CI = −0.21–0.85, *p* = 0.24) with the heterogeneity among the studies being I^2^ = 0% at *p* = 0.99 (Figure 3).

The meta-analysis results for balance in the post-treatment CIMT group were unable to display a significant difference in balance when compared to the control groups; the SMD was 4.94 at 95% CI = −2.48–12.67, *p* = 0.19, with substantial heterogeneity among the studies (I^2^ = 92% at *p* < 0.001). In addition, in studies reporting follow-up of treatment sessions, the CIMT group was unable to display a significant difference in balance when compared to the control groups (SMD = 3.84 at 95% CI = −2.33–10.01, *p* = 0.22); the heterogeneity among the studies was I^2^ = 88% at *p* = 0.004 (Figure 4).

## 4. Discussion

This review discusses the qualitative and quantitative effects of LECIMT on the numerous parameters that determine functional ambulation. The current review is unique compared to previous reviews because it primarily observed gait and balance cardiopulmonary parameters exclusively among RCTs. The gait outcome measures were extensively reported in seven studies [50,51,52,54,55,56,58,59], followed by balance in four studies [53,56,58,59], cardiovascular fitness [50], functional mobility [53], functional ambulation [59], and motor functions [52]. In two trials, LECIMT showed superior effects for outcome measures when compared to its respective control group [58,59]. Eight studies exhibited an improvement in post and follow-up sessions similar to the control group for all outcome measures [48,51,52,54,55,56], except for cardiovascular parameters, where there were no clear changes from pre- to post-sessions [50]. Similar findings in earlier reviews support the positive effects of LECIMT on gait speed, balance, functional mobility, functional ambulation, and motor functions [60]. However, compared to the control group, LECIMT protocols exhibited ineffectiveness in improving the speed of walking and functional balance in the present meta-analysis.

We found equivalent or better results in the experimental groups, including all or some of the core principles (intensive practice with repetitions, shaping, and transfer package) of CIMT. The beneficial effects of restraining a non-paretic upper limb in ULCIMT are due to the opportunity of engaging a paretic limb independently in functional task practice. In contrast, lower-extremity functions are typically bipedal, resulting in the paretic limb being used forcibly during LECIMT. This issue may explain why there were similar influences on the outcomes of the included studies regardless of the application of constraints.

Even though we expect the neuroplastic changes within six months following a stroke to enhance functional recovery [61], post-six months, functional improvements are not impossible [62]. Moreover, after six months of post-stroke intense practice simulating practical tasks, functional improvements are inevitable in stroke patients [42,63]. However, we observed that many studies did not include the core principles of CIMT, including shaping, hence emphasizing the progressive modifications among the treatment parameters, such as number, type, and complexity of the tasks [64,65,66]. Typically, therapists withhold treatment training due to a plateau in functional improvements in chronic stroke patients; however, with CIMT treatment, strategies including repetitive high-intensity training could produce neuroplasticity and functional improvements even after reaching a plateau in their improvements [67,68,69,70]. Many studies designed to maintain the same level of task training without progressing to the next level of challenge would have failed to achieve any better results than the conventional practice.

Lower-extremity motor recovery is more complex and delayed in stroke populations. Community ambulation is an integral component in improving quality of life (QOL), for which therapists require gait training, including multi-tasking, obstacle training, and training on different surfaces rather than simple task-specific training [42,63]. Some studies showed further improvements through LECIMT by emphasizing that stroke patients should perform multiple functional tasks at home and by promoting social support, self-monitoring, and motivation for better adherence as a part of a “transfer package” [56,59].

The predictable walking ability, functional capacity, and QOL of stroke patients are strongly associated with cardiovascular efficiency [6,71,72]. Variations in blood pressure are an essential indicator of the possibility of hypertension in stroke, cardiovascular disease, high risk of recurrent stroke, renal failure, and all-cause mortality [73]. Hence, blood pressure variability should be used in clinical trials to evaluate the efficacy of treatment against cardiovascular-related mortality in stroke [73].

Promoting low-to-moderate physical activity is essential in stroke risk management. Moreover, aerobic exercise regimes increase the production of new blood vessels, nerve cells, and synaptic connections [74,75]. In the initial phases of the stroke, muscle strength and balance play a key role in determining functional ambulation. In later stages, cardiovascular capacity plays a significant role along these factors [23]. Despite being an essential component in stroke recovery, most studies included in the analysis surprisingly missed it as an outcome measure.

In this review, most studies preferred treadmills for gait training, improving cardiovascular capability. However, most studies missed setting the target level of intensity for cardiovascular conditioning and its progression, which is a crucial factor in improving cardiovascular fitness [76,77,78,79].

The gait speed and balance meta-analysis did not show any significant effect size. The type of gait training prescribed in experimental and control groups, heterogeneity in the restraint used, unequal treatment durations, small sample size, chronicity of the stroke, a smaller number of studies, and low methodological quality were to be credited for insignificant differences between the groups.

CIMT demands highly specific training for its efficient usage in clinical practice. However, therapists require expertise in identifying individuals’ functional needs, determining and framing the functionally related tasks for practice, setting targets, progressing the intensity, monitoring improvements, and applying CIMT principles. We suggest that the in-expertise in the application of CIMT might contribute to similar improvements in both groups [42].

In some of the studies, many authors did not demonstrate improvements due to inefficient usage of the principles of CIMT, such as transfer package and shaping, lack of feedback, absence of functional task training, and not progressively increasing task complexity. Along with this involvement of various stroke populations with multiple phases and severities of stroke, variability in treatment durations, the intensity of training, and the type of tasks might be the reasons for equal effects in their counterparts.

The included RCTs could have been improved by providing more appropriate sample sizes and homogeneity in their intervention plans. In addition, most RCTs did not include various important outcome measures, such as QOL, functional mobility, and cardiovascular parameters. Due to the scarcity of RCTs, we included some RCTs with low methodological quality and with a single session of training. Moreover, further meta-analysis for other gait parameters, mobility and balance variables of lower-limb-related functions, and QOL could be warranted if sufficient RCTs are available.

## 5. Conclusions

In this meta-analysis, we observed that despite the fact that several trials claimed the efficacy of LECIMT in improving lower-extremity functions, gait speed and balance did not demonstrate a significant effect size favoring LECIMT. Moreover, forthcoming CIMT treatment protocols should consider the functional requirements of stroke subjects and the appropriate application of all its principles.

## Figures and Tables

**Figure 1 ijerph-19-12809-f001:**
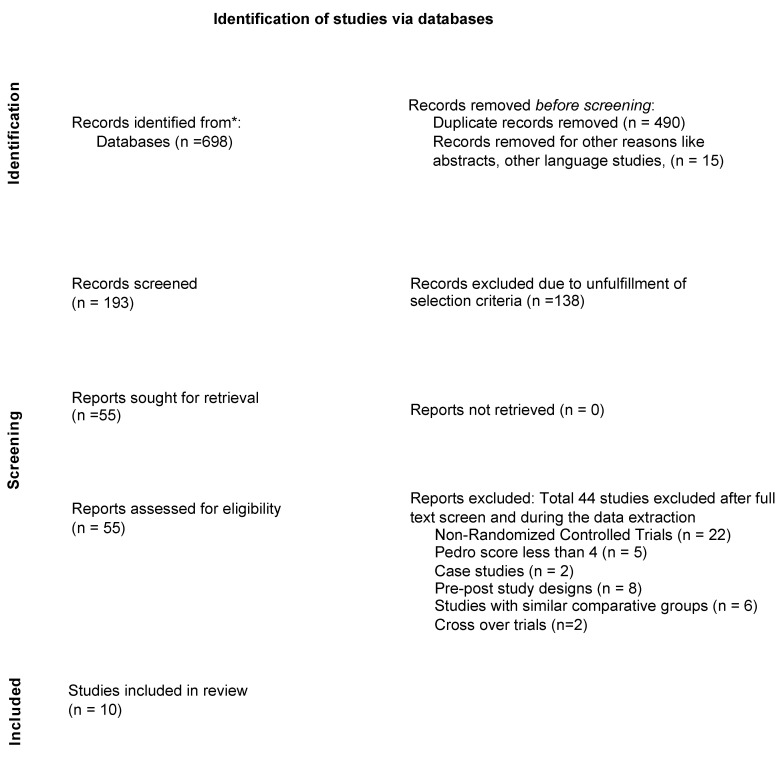
Flowchart depicting the process of synthesis of included studies for this review.

**Figure 2 ijerph-19-12809-f002:**
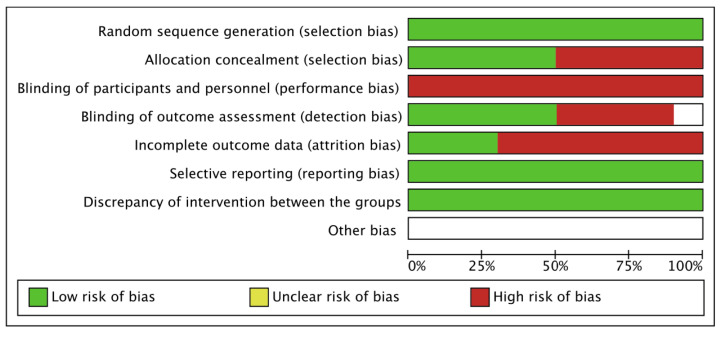
Details of risk of bias among the included studies.

**Figure 3 ijerph-19-12809-f003:**
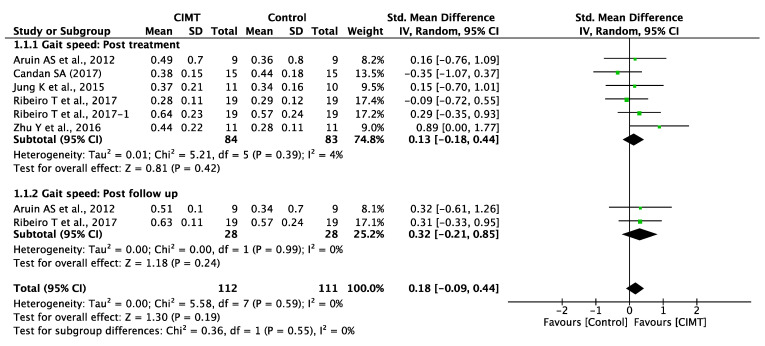
Gait speed: post-treatment and post-follow-up.

**Figure 4 ijerph-19-12809-f004:**
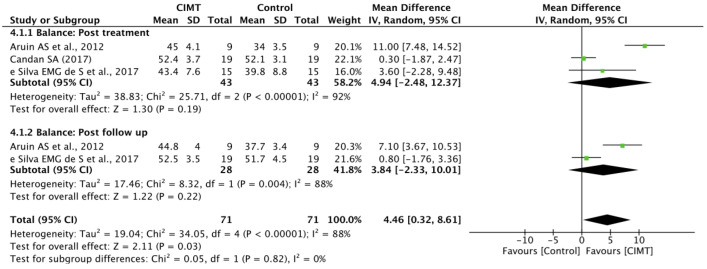
Balance: post-treatment and post-follow-up.

**Table 1 ijerph-19-12809-t001:** Search strategy utilized in the study.

Databases	PICO Format Search with Bullion Keywords (And) (OR)			
	Patient	Intervention	Comparison	Outcome
EBSCO, PubMed, PEDro, Science Direct, Scopus, MEDLINE, CINAHL, and Web of Science	Stroke OR Hemiplegia OR Hemiparesis OR Cerebrovascular accident	CIMT OR Constraint Induced Movement therapy OR Restricted Limb/Extremity OR Forced use	Proprioceptive Neuromuscular Facilitation OR PNF OR Neuro-Developmental Treatment R NDT OR Conventional Physical Therapy OR CPT OR Physiotherapy OR Exercise OR Or traditional rehabilitation OR Standard Physical Therapy	Gait speed OR Gait velocity OR Balance OR Center of Gravity OR Base of Support OR Center of Pressure OR Cardiovascular parameters OR Blood pressure OR percentage of heart rate maximum.

**Table 2 ijerph-19-12809-t002:** Characteristics of studies on lower-extremity constraint-induced movement therapy (LECIMT).

Author/Year	Age	Chronicity	Intervention			Outcome Measures	Inferences
			Experimental	Control	Duration		
Aruin AS et al., 2012	57.7 ± 11.9	Chronic	A shoe insert is provided on the unaffected side to shift body weight onto the affected side to promote muscle strength and weight-bearing capability.	The treatment encompasses the promotion of weight-bearing towards the affected side to promote balance and muscle strength.	60 min per session, one session per week, six sessions in total, 6 h.	Symmetrical weight bearing, gait speed (m/s), BBS, Fugl-Meyer for lower extremity.	Post and follow-up retention were observed in the experimental group for symmetrical weight bearing, gait speed, and BBS in the experimental group.
Bonnyaud C et al., 2013	50.03 ± 13.1	Chronic	Treadmill training with ankle mass on non-paretic lower extremity.	Treadmill training.	20 min, single session.	Cadence (steps/min), step length (cm), peak hip and knee flexion and dorsiflexion, vertical GRF (N/Kg), peak propulsion (N/Kg), peak breaking (N/Kg) gait speed (m/s).	The experimental and control group showed similar effects for gait variables.
Bonnyaud C et al., 2014	50.6 5 ± 11.65	Chronic	Asymmetrical gait training group: RAGT providing negative kinematic restraint applied to non-paretic lower extremity.	Symmetrical RAGT gait-training group.	20 min, single session.	Symmetry ratio, stance time, double support time, static and dynamic GRF.	Peak knee flexion range was improved in the asymmetrical robotic raining group, and other gait variables improved equally among symmetrical and asymmetrical RAGT groups.
Jung K et al., 2015	56.35 ± 14.1	Subacute/chronic	Auditory feedback provided while walking with a cane in addition to standard therapy.	Walking with a cane in addition to standard therapy.	Gait training: 5 days per week for four weeks, 30 min per session. Standard therapy: Five days per week for four weeks, 30 min per session.	Gluteus medius and vastus medialis oblique muscle activity, single support phase of the affected side (% GC) vertical peak force of the cane (% BW) and gait speed (m/s).	The experimental group showed significant improvement in muscle activation and gait speed.
Zhu Y et al., 2016	58.71 ± 6.02	Subacute	Gait training consists of 2 h of sit-to-stand transfers, indoor walking, climbing up and down stairs, balance training and one-leg weight training with more repetitions in addition to this standard comprehensive rehabilitation.	Standard comprehensive rehabilitation treatment includes passive and active ROM exercises, stretching, balance and gait training.	Four weeks five days per week.	Step length (m), COM displacements, swing time (%gait cycle) step width(m), and gait speed(m/s).	m-CIMT gait training improved both COM displacements and spatio-temporal gait parameters.
Ribeiro T et al., 2017	57.75 ± 3.75	Subacute/chronic	Gait training on a treadmill, applying weight on the unaffected side.	Gait training on a treadmill.	The nine training sessions, 30 min, two consecutive weeks.	Step length, hip, knee and ankle ROM, and gait speed(m/s).	Spatio-temporal and kinematic gait parameters improved in both groups equally.
e Silva EMG de S et al., 2017	57.75 ± 3.75	Subacute/chronic	Gait training on a treadmill, applying weight on the unaffected side.	Gait training on a treadmill.	The nine training sessions, 30 min, two consecutive weeks.	BBS, stride time(s), TUG, symmetry ratio, stride width(m), turn speed(m/s), and stride length(m).	Spatio-temporal gait parameters balance and functional mobility improved in both groups equally.
Candan SA et al., 2017	56.4 ± 13.45	Subacute/chronic	m-CIMT includes intensive practice, restrain of non-paretic lower extremity and transfer package.	NDT program.	120 min per session, five sessions per week for two weeks.	BBS, step length ratio, cadence (steps/min), postural symmetry FAC, and gait speed.	The m-CIMT group showed significant improvements on all variables when compared to the NDT group.
Ribeiro T et al., 2017	57.75 ± 3.75	Subacute/chronic	Gait training on a treadmill, applying weight on the unaffected side.	Gait training on a treadmill.	30 min per session, nine training sessions for two consecutive weeks.	SPB (mmHg), DPB (mmHg), % of HR max, distance covered (m), gait speed (m/s).	Kinetic gait parameters improved in both groups equally. Restraint of a non-paretic limb did not show any effect. No changes have been observed in cardiovascular parameters between pre and post sessions.
Ribeiro T et al., 2020	57.75 ± 3.75	Subacute/chronic	Gait training on a treadmill, applying weight on the unaffected side.	Gait training on a treadmill.	The nine sessions, 30 min, two consecutive weeks.	Stance time(s), static and dynamic (GRF), double support time (s), symmetrical weight bearing, and symmetry ratio.	The experimental and control group showed similar effects for gait variables.

Notes: BBS: Berg balance scale; (m/s): (meters/second); (m): (meters); (cm): centimeters, GRF: ground reaction force; (N/Kg): (newtons/kilogram); (% GC): percentage of gait cycle; (% BW): percentage of body weight; ROM: range of motion; TUG: time up and go test; FAC: functional ambulation category; SPB: systolic blood pressure; DPB: diastolic blood pressure; % of HR max: percentage of heart rate maximum; RAGT: robotic-assisted gait training; NDT: neuro-developmental therapy; m-CIMT: modified constraint-induced movement therapy.

**Table 3 ijerph-19-12809-t003:** Quality assessment for randomized control trials (RCTs) using Physiotherapy Evidence Database (PEDro) scale.

Study ID	Eligibility Criteria	Random Allocation	Concealed Allocation	Baseline Comparability	Blinding of Participants	Blinding of Therapist	Blinding of Assessor	Adequate Follow-Up (>85%)	Intention to Treat	Between-Group Comparison	Point Estimates and Variability	Pedro Score (10)
Aruin AS et al., 2012	Y	Y	N	Y	N	N	N	×	N	Y	Y	4
Bonnyaud C et al., 2013	N	Y	N	Y	N	N	N	N	N	Y	Y	4
Bonnyaud C et al., 2014	N	Y	N	Y	N	N	N	N	N	Y	Y	4
Jung K et al., 2015	Y	Y	Y	Y	N	N	Y	Y	N	Y	Y	7
Zhu Y et al., 2016	Y	Y	N	Y	N	N	Y	N	N	Y	Y	5
Ribeiro T et al., 2017	Y	Y	Y	Y	N	N	Y	Y	Y	Y	Y	8
e Silva EMG de S et al., 2017	Y	Y	Y	Y	N	N	Y	Y	Y	Y	Y	8
Candan SA et al., 2017	Y	Y	N	Y	N	N	Y	Y	N	Y	Y	6
Ribeiro T et al., 2017	N	Y	N	Y	N	N	N	Y	N	Y	Y	5
Ribeiro T et al., 2020	Y	Y	Y	Y	N	N	Y	Y	Y	Y	Y	8

Notes: “Y”: yes; “N”: no.

**Table 4 ijerph-19-12809-t004:** Level of evidence for outcome measures included in the review.

Outcome Measures		Level of Evidence	Quality of the Studies
Gait parameters	Gait speed	Level 1b	Good
Cardiovascular parameters	SPB (mmHg), DPB (mmHg), % of HR max	Level 2	Fair
Balance	BBS and Postural symmetry	Level 1b	Good

Notes: BBS: Berg balance scale; SPB: systolic blood pressure; DPB: diastolic blood pressure; % of HR max: percentage of heart rate maximum.

## Data Availability

This study did not report any data.

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
