# Peer review of "Impact of Constraint-Induced Movement Therapy (CIMT) on Functional Ambulation in Stroke Patients—A Systematic Review and Meta-Analysis"

_ijerph, 2022, doi:10.3390/ijerph191912809_

Round 1

Reviewer 1 Report

This meta-analysis review paper compares the RCTs with regard to the effect of LECIMT physiotherapy on the balance and gait speed for the stroke patients.

Methods:

L. 136: PEDro score has not been explained. What does it show? Is there any quality cutoff which was used to select the studies? I believe this is something that has to be mentioned in the Methods section as well as the meaning and scale of the PEDro scoring system.

L. 154: I2 statistic is known to be less reliable for the small number of studies in the meta-analysis (only 10 studies here). How do the authors mitigate this?

For example as discussed here:

von Hippel, P.T. The heterogeneity statistic I2 can be biased in small meta-analyses. BMC Med Res Methodol 15, 35 (2015). https://doi.org/10.1186/s12874-015-0024-z

Further in the text it becomes evident that the authors also conducted other statistical tests, but not properly described in this section or further in the paper. To facilitate the reproducibility of this study, it is also important to state what software/programming language (version) was used to conduct these analyses.

L. 155: "...treatment effects of each outcome.." The outcomes are balance and gait speed? probably worth mentioning them here again.

Results:

Figure 1: is confusing. It starts from 692 citations, then the following box says 886 non-duplicate citation were looked at. I would expect the number of citations in the "child" box to be equal or smaller than in the "parent" box.

Why tables with the results are shown as figures (screenshots perhaps)? The contents are too small and difficult to see. I would recommend recreating these tables in any suitable software (whichever was used to produce this paper such as MS Word or LaTeX) to improve the readability. They also do not fit the text borders.

Many more outcomes were evaluated in the mentioned RCTs, but only the effect on balance and gait speed was considered in this meta-analysis. It does not come across clearly to me why the focus is specifically on these outcomes and not the others?

It would be useful to know in the clinical practice which outcomes have shown a consistent improvement as a result of intervention and which not across the studies, and whether there are any consistent or statistically significant improvements for other outcomes.

The paper also mentions cardiovascular outcomes in the abstract, but no statistical results seem to be present for those outcomes in the main text.

Minor corrections:

L. 80: perhaps, "The application of constraint to.."

L. 89, 97: not sure what "task shaping" means in this context. Maybe more explanation here for the readers who are not familiar with CIMT would be helpful?

L. 112: "LECMT" -> "LECIMT"

L. 133: maybe "..were searched for the relevant publications"

L. 154: "I2 statistics" -> "I2 statistic"

Figure 1: "trail" -> "trial"

L. 188: What does RAGT stand for?

L. 201: "Table 1 summarises the details.. "

Table 1: Age 57.7 +- 311.9? Mistyping?

             50.03 13.1? +- is missing I guess

This table should be checked again for the word duplicates and misspellings.

L. 214: "on unaffected and paretic..": limb?

L. 216: GRF?

L. 250: PEDro scores perhaps? References not properly formatted as well

Author Response

Response to reviewer comments

Thank you so much for your valuable time to review and for considering our study. The suggestions offered by the reviewers were immensely helpful. We carefully considered changes in the manuscript as per your suggestions. We modified and highlighted the changes in the manuscript. The highlighted changes are in green color. Here is a point-by-point explanation of reviewer comments. Thank you once again.

Sl.no

Queries

Response to queries

1.      

L. 136: PEDro score has not been explained. What does it show? Is there any quality cutoff which was used to select the studies? I believe this is something that has to be mentioned in the Methods section as well as the meaning and scale of the PEDro scoring system.

As per the recommendations Pedro scale was explained in detail. The quality cutoff for selection of studies to include in the review mentioned in Line: 168 of methodology section.

2.      

L. 154: I2 statistic is known to be less reliable for the small number of studies in the meta-analysis (only 10 studies here). How do the authors mitigate this?

For example as discussed here:

Von Hippel, P.T. The heterogeneity statistic I2 can be biased in small meta-analyses. BMC Med Res 3Methodol 15, 35 (2015). https://doi.org/10.1186/s12874-015-0024-z

This is the valuable point raised by the reviewer. As you know as of now the I2 statistics are most done in health-related meta-analysis. The authors were careful about their conclusions in this study, along with the results of this metanalysis the authors considered the conclusions provided by individual authors. The individual study results were also observed to see whether the statistical improvements provided in the studies are clinically relevant and sufficient. By careful understanding of the reference provided by you we have reviewed our findings and it was immensely helpful for the authors to mitigate the errors.

3.      

Further in the text it becomes evident that the authors also conducted other statistical tests, but not properly described in this section or further in the paper. To facilitate the reproducibility of this study, it is also important to state what software/programming language (version) was used to conduct these analyses

As per the recommendations the prescribed software were mentioned for all the statistical tests in methodology sections

4.      

L. 155: "...treatment effects of each outcome.” The outcomes are balance and gait speed? Probably worth mentioning them here again.

As per the recommendation outcomes gait speed and balance were mentioned at line:198

5.      

Figure 1: is confusing. It starts from 692 citations, then the following box says 886 non-duplicate citation were looked at. I would expect the number of citations in the "child" box to be equal or smaller than in the "parent" box.

As per the recommendations we accepted our error, and the details of search strategy was rectified in figure1

6.      

Why tables with the results are shown as figures (screenshots perhaps)? The contents are too small and difficult to see. I would recommend recreating these tables in any suitable software (whichever was used to produce this paper such as MS Word or LaTeX) to improve the readability. They also do not fit the text borders.

The figures which represent the results of metanalysis for gait speed and balance were obtained from the Review manager 5.4.1 software. We depicted them as we attained from the software to maintain the originality. As per the suggestions we tried to convert it into MS Word or LaTeX but, alignment of the figure is getting distorted. So we tried our level best to make it clear

7.      

Many more outcomes were evaluated in the mentioned RCTs, but only the effect on balance and gait speed was considered in this meta-analysis. It does not come across clearly to me why the focus is specifically on these outcomes and not the others?

 As per the protocol registration our primary outcome measures of interest were gait speed, balance, and cardiovascular parameters.  We selected the above outcome measures as they are major determinants of functional ambulation and the same was mentioned in the introduction. We are sorry for the confusion created by us with so many outcome measures which are not in scope. We modified our outcome measures in the manuscript for appropriateness

8.      

It would be useful to know in the clinical practice which outcomes have shown a consistent improvement as a result of intervention and which not across the studies, and whether there are any consistent or statistically significant improvements for other outcomes

As per the recommendations we have mentioned the improvements reported for each outcome measures across the included studies in the 3.3: outcome measures subsection  of section 3: results

9.      

The paper also mentions cardiovascular outcomes in the abstract, but no statistical results seem to be present for those outcomes in the main text.

We have mentioned cardiovascular parameters by mistake as cardiorespiratory parameters. As cardiovascular parameters were reported in only one study, we couldn’t able to perform statistical analysis

10.   

L. 80: perhaps, "The application of constraint to.”

We corrected it as “constraint” as it is appropriate term to be used .

11.   

L. 89, 97: not sure what "task shaping" means in this context. Maybe more explanation here for the readers who are not familiar with CIMT would be helpful?

As per the suggestion we defined the “task shaping” through line 91-93 in the introduction

12.   

L. 112: "LECMT" -> "LECIMT"

LECMT corrected as LECIMT

13.   

L. 133: maybe ".were searched for the relevant publication

Corrected as suggested

14.   

L. 154: "I2 statistics" -> "I2 statistic"

Corrected as suggested in

15.   

Figure 1: "trail" -> "trial

We changed figure: 1 for better understanding

16.   

L. 188: What does RAGT stand for

RAGT stand for “Robotic assisted gait training”.  Changes incorporated in the manuscript.

17.   

L. 201: "Table 1 summarizes the details."

Corrected as suggested

18.   

Table 1: Age 57.7 +- 311.9? Mistyping?

50.03 13.1? +- is missing I guess

Corrected the mistyping in Table 1

19.   

This table should be checked again for the word duplicates and misspellings.

As per the recommendation we thoroughly checked and corrected word duplicates and spellings

20.   

L. 214: "on unaffected and paretic.” limb?

The line was modified in the manuscript.

21.   

L. 216: GRF?

The line was modified in the manuscript

GRF: Ground reaction force

22.   

L. 250: PEDro scores perhaps? References not properly formatted as well

As per the recommendation, we formatted the references.

Reviewer 2 Report

This systematic review and meta-analysis describes the effect of various lower extremity CIMT (LECIMT) protocols on gait speed, balance, and cardiovascular outcomes. This review is of clinical interest and the methodology used is adequate. The introduction describes the rationale for the review in the context of what is already known. The methods describe the information of the databases used, the process of selecting the studies and extracting the results. The results are well reported in tables and forest plots. In the discussion are summarized the main findings and a general interpretation of the results in the context of other evidence.

However, there are also some crucial limitations:

-          The manuscript is not reported according to the PRISMA reporting guideline for systematic reviews. Therefore some relevant information is omitted.

-          This review has not been prospectively registered on a public register, such as PROSPERO. This procedure is necessary when performing systematic reviews and meta-analyses.

-          It is not specified if the eligibility criteria for considering studies were based on PICOS elements (i.e. participants, intervention, comparators, outcomes, study design).

-          The methodological quality of the studies using the PEDRO scale has been reported, however individual and global risk of bias has not. It is recommended to use the Cochrane risk of bias tool for randomized trials (RoB 2).

-          Taken into consideration that the outcomes are spatio-temporal gait variables, kinematic gait variables, kinetic gait variables, balance, cardiorespiratory parameters, functional ambulation and lower extremity function, why only quantitative results of gait speed and balance are described?

-          Since there is a systematic review on this topic, it would be recommendable to go into more detail on why the present review is important, or some remarkable differences. (Abdullahi, A.; Truijen, S.; Umar, NA; Useh, U.; Egwuonwu, V.A.; Van Criekinge, T.; Saeys, W. Effects of Lower Limb 512 Constraint Induced Movement Therapy in People With Stroke: A Systematic Review and Meta-Analysis. Front. Neurol. 2021, 513 12, 343.)

Other comments:

-          The Data analysis section does not specify which software has been used for the metaanalysis.

-          It is recommended to extend the search limit to the current date (2022).

-          The flow chart (Figure 1) is missing information. The databases used were Google Scholar, PEDro, Web of Science, PubMed, Scopus EBSCO, and Science Direct, however, only Web of Science is included in the figure. In addition, the numbers increase from 602 citations to 886 citations, and the reasons for exclusion are specified for only 12 studies out of 44.

-          Some formatting issues, e.g., number 2 in superscript for I (I2), or some references without brackets.

-          In the first paragraph of the discussion, check reference 48 and 50, which of the two reports results on cardiorespiratory fitness?

Author Response

Response to reviewer comments

Reviewer 2

Sl.no

Queries

Response to queries

This systematic review and meta-analysis describe the effect of various lower extremity CIMT (LECIMT) protocols on gait speed, balance, and cardiovascular outcomes. This review is of clinical interest and the methodology used is adequate. The introduction describes the rationale for the review in the context of what is already known. The methods describe the information of the databases used, the process of selecting the studies and extracting the results. The results are well reported in tables and forest plots. In the discussion are summarized the main findings and a general interpretation of the results in the context of other evidence.

Thank you so much for your valuable time to review and for considering our study. The suggestions offered by the reviewers were immensely helpful. We carefully considered changes in the manuscript as per your suggestions. We modified and highlighted the changes in the manuscript. The highlighted changes are in yellow color. Here is a point-by-point explanation of reviewer comments. Thank you once again

1

The manuscript is not reported according to the PRISMA reporting guideline for systematic reviews. Therefore some relevant information is omitted.

As per the suggestions we reported the manuscript according to the PRISMA guidelines

2

This review has not been prospectively registered on a public register, such as PROSPERO. This procedure is necessary when performing systematic reviews and meta-analyses.

As per the suggestions we specified the PROSPERO registration number which was prospectively registered

3

It is not specified if the eligibility criteria for considering studies were based on PICOS elements (i.e. participants, intervention, comparators, outcomes, study design).

As per the suggestion we specified the eligibility criteria considering the PICO format

4

The methodological quality of the studies using the PEDRO scale has been reported, however individual and global risk of bias has not. It is recommended to use the Cochrane risk of bias tool for randomized trials (RoB 2).

As per the suggestion we have included Cochrane risk of bias tool

5

Taken into consideration that the outcomes are spatio-temporal gait variables, kinematic gait variables, kinetic gait variables, balance, cardiorespiratory parameters, functional ambulation and lower extremity function, why only quantitative results of gait speed and balance are described?

As per the recommendation we have considered other parameters also for analysis binding to the requirements of metanalysis.

6

Since there is a systematic review on this topic, it would be recommendable to go into more detail on why the present review is important, or some remarkable differences. (Abdullahi, A.; Truijen, S.; Umar, NA; Useh, U.; Egwuonwu, V.A.; Van Criekinge, T.; Saeys, W. Effects of Lower Limb 512 Constraint Induced Movement Therapy in People With Stroke: A Systematic Review and Meta-Analysis. Front. Neurol. 2021, 513 12, 343.)

As per the recommendation we referred to the previous metanalysis and specified the importance of present review

1.     The previous systematic review included randomized control trails, pre-post study designs, case studies, and randomized control trails prescribing same intervention in both groups which may affect the quality of the study, and its interpretations should be carefully considered.  To overcome the previous studies drawbacks and to provide high level of evidence the present systematic review considered exclusively randomized control trails with PEDro score more than 4.

2.     The present review considered major components of functional ambulation such as gait speed, balance, and cardiovascular parameters. In the previous review cardiovascular parameters was not studied

3.     The number of studies considered for gait speed metanalysis in the present review were 6 whereas in the previous review only 3 studies were included which may hugely affects the treatment effect size

7

Line 89: Google Scholar more than a database, is considered gray literature. Please specify databases, rather than search engines.

As per the recommendations we modified the databases.

8

The Data analysis section does not specify which software has been used for the meta-analysis.

As per the recommendation we specified about software used for the meta-analysis. Line no

9

It is recommended to extend the search limit to the current date (2022).

As per the recommendation we extended the search till date and same was specified in line no

10

The flow chart (Figure 1) is missing information. The databases used were Google Scholar, PEDro, Web of Science, PubMed, Scopus EBSCO, and Science Direct, however, only Web of Science is included in the figure. In addition, the numbers increase from 602 citations to 886 citations, and the reasons for exclusion are specified for only 12 studies out of 44.

As per the recommendation we reevaluated and corrected the number of citation and furnished the details about excluded studies in the flowchart (Figure-1)

11

Some formatting issues, e.g., number 2 in superscript for I (I2), or some references without brackets.

As per the suggestion the formatting issues and references were corrected

12

In the first paragraph of the discussion, check reference 48 and 50, which of the two reports results on cardiorespiratory fitness?

As per your recommendation we revised and corrected the reference

Round 2

Reviewer 1 Report

L.135-136: the sentence better to be rephrased as its somewhat complicated, "PEDro, Web of Science, ... databases were searched for the relevant articles."

L. 165-167 the sentence is repeated in L. 169-170. The second one better to be removed.

L. 165: "RCTs which scored...", L. 170: "RCTs with a score of..."

Table 1 has been substantially improved, but still a few mistypings/misspellings here and there. For example, "asymmetrical robotic raining group", I think should be "training group". I suggest checking it one more time, but generally is acceptable.

One minor note is the capital letters often used in the sentences, but don't seem to be appropriate for the word or the context, proofreading would help.

Author Response

Response to reviewer comments

Thank you so much for your valuable time reviewing and considering our study. The suggestions offered by the reviewers were constructive. We carefully thought changes in the manuscript as per your suggestions. As a result, we modified and highlighted the changes in the manuscript. The highlighted changes are in green colour. Here is a point-by-point explanation of reviewer comments. Thank you once again.

Sl.no

Queries

Response to queries

1.      

L.135-136: the sentence better to be rephrased as its somewhat complicated, "PEDro, Web of Science, ... databases were searched for the relevant articles."

As per the recommendations, we rephrased the sentence

2.      

L. 165-167 the sentence is repeated in L. 169-170. The second one better to be removed.

As per the recommendations, we removed the sentence ws repeated

3.      

L. 165: "RCTs which scored...", L. 170: "RCTs with a score of..."

As per the recommendations, we rephrased the sentence

4.      

Table 1 has been substantially improved, but still a few mistypings/misspellings here and there. For example, "asymmetrical robotic raining group", I think should be "training group". I suggest checking it one more time, but generally is acceptable.

As per the suggestion, Table 1 was rechecked for mistyping /misspellings, and the same was corrected.

5.      

One minor note is the capital letters often used in the sentences, but don't seem to be appropriate for the word or the context, proofreading would help.

The manuscript is edited by Cambridge proofreading services. The English editing certificate is attached.  

Reviewer 2 Report

The authors have taken into account the comments and suggestions proposed in the previous review.

Author Response

Response to reviewer comments

Sl.no

Queries

Response to queries

1

The authors have taken into account the comments and suggestions proposed in the previous review.

Thank you for your time and effort.
